Waterlogging tolerance and recovery capability screening in peanut: a comparative analysis of waterlogging effects on physiological traits and yield

Zeng Ruier
Cao Jing
Li Xi
Wang Xinyue
Wang Ying
Yao Suzhe
Gao Yu
Hu Jing
Luo Mingzhu
Zhang Lei zhanglei@scau.edu.cn
Chen Tingting chentingting@scau.edu.cn
College of Agriculture, South China Agricultural University , Guangzhou , Guangdong , China
Hasanuzzaman Mirza
Electronic publication date: 2022 Jan 12
Publication date: 2022
Volume: 10
Electronic Location ID: e12741
Received 2021 Aug 16; Accepted 2021 Dec 13
Copyright: ©2022 Zeng et al.
Copyright year: 2022
Copyright holder: Zeng et al.
License: This is an open access article distributed under the terms of the Creative Commons Attribution License, which permits unrestricted use, distribution, reproduction and adaptation in any medium and for any purpose provided that it is properly attributed. For attribution, the original author(s), title, publication source (PeerJ) and either DOI or URL of the article must be cited.
License URL: https://creativecommons.org/licenses/by/4.0/

Keywords: Arachis hypogaea L., Waterlogging stress, Varities, Comprehensive evaluation, Waterlogging tolerance, Recovery capability

Funding: The National Key R&D Program of China 2020YFD1000905 Key Science and Technology Planning Project of Guangdong Province 2019B020214003 Guangdong Technical System of Peanut and Soybean Industry 2019KJ136-05 This work was supported by the National Key R&D Program of China (2020YFD1000905), the Key Science and Technology Planning Project of Guangdong Province (2019B020214003), and the Guangdong Technical System of Peanut and Soybean Industry (2019KJ136-05). The funders had no role in study design, data collection and analysis, decision to publish, or preparation of the manuscript.

==============================
Fifteen peanut varieties at the pod filling stage were exposed to waterlogging stress for 7 days, the enzyme activities and fluorescence parameters were measured after 7 days of waterlogging and drainage. The waterlogging tolerance and recovery capability of varieties were identified. After waterlogging, waterlogging tolerance coefficient (WTC) of relative electrolyte linkage (REL), malondialdehyde (MDA) content, superoxide dismutase (SOD) activity, and catalase (CAT) activity, non-photochemical quenching (NPQ) and photochemical quenching (qL) of leaves of most peanut varieties were increased, while the WTC of the soil and plant analysis development (SPAD) value, PS II actual quantum yield (ΦPS II), maximum photochemical efficiency (Fv/Fm) were decreased. After drainage, the WTC of REL, MDA content, SOD and CAT activity of leaves were decreased compared with that of after waterlogging, but these indicators of a few cultivars were increased. ΦPS II, Fv/Fm and qL can be used as important indexes to identify waterlogging recovery capability. There was a significant negative correlation between recovery capability and the proportion of reduction in yield, while no significant correlation was found between waterlogging tolerance and the proportion of reduction in yield. Therefore, it is recommended to select varieties with high recovery capability and less pod number reduction under waterlogging in peanut breeding and cultivation.

Introduction

Waterlogging is one of the limiting factors affecting plant growth and development, resulting in a sharp yield decline and huge economic losses (Bailey-Serres, Lee & Brinton, 2012). Waterlogging causes hypoxia in plant roots, and acetaldehyde, ethanol and other substances produced by anaerobic metabolism in roots are toxic to root cells, which inhibits carbon assimilation and photosynthate utilization (Aydogan & Turhan, 2015; Gao et al., 2021). Besides, waterlogging breaks the dynamic balance of plant reactive oxygen species production and scavenging (Hu et al., 2020), resulting in the accumulation of reactive oxygen radicals in plants and membrane lipid peroxidation and cell dysfunction (Liu et al., 2010; Wei et al., 2013). Waterlogging also resulted in a decrease in photosynthetic rate, leaf yellowing and wilting. Therefore, plant growth is inhibited and eventually leads to the decrease of total biomass and yield (Tian et al., 2021; Zhang et al., 2019).

Peanut (Arachis hypogaea L.) is a widely planted leguminous crop in the world. As an important source of oil and protein for human beings, peanut has high nutritional value and a wide range of uses (Bishi et al., 2015; Zhao et al., 2019). In China, due to excessive rainfall, the peanut field was flooded, which severely restricted the peanut production (Zeng et al., 2020). Previous studies have revealed that the photosynthetic system of the waterlogged peanut leaves was destroyed, limiting the CO2 assimilation rate and reducing the photosynthetic efficiency of peanut leaves. The most susceptible growth stage of peanut to waterlogging was the pod filling stage, and waterlogging at the pod filling stage significantly reduced the pods number per plant and pod weight, and ultimately led to peanut yield decline (Zeng et al., 2020; Bishnoi & Krishnamoorthy, 1992). With global warming, the growth and development of peanut plants are facing an increasing risk of waterlogging (Schiermeier, 2011). Therefore, identifying and screening peanut varieties with high waterlogging tolerance has become a critical problem to be addressed urgently in the peanut production areas of China.

As we know, the waterlogging tolerance of peanut is a complex and comprehensive trait, so it is significant to develop an effective method to screen waterlogging tolerant peanut varieties. The recuperative potential from stress is of great significance for plant vitality and survival (Galle, Haldimann & Feller, 2007; Shi et al., 2016). Combining the physiological and growth characteristics of various varieties after waterlogging and during the recovery stage is an effective method to identify the waterlogging tolerance of different varieties (Ciancio et al., 2021; Aydogan & Turhan, 2015; Pompeiano et al., 2019). A study on mungbean suggested that selecting the genotype with the least decrease in chlorophyll fluorescence after drainage was beneficial for screening a large number of waterlogging tolerant varieties (Aydogan & Turhan, 2015). Besides, the ability to conserve water content and high photosynthetic capacity through stomatal control was important for the regrowth of kiwifruit vines (Li et al., 2020). Meanwhile, in agricultural production, only the maximum yield can ensure the output of farmland and the income of farmers. However, waterlogging affects the yield and yield components. For example, waterlogging reduced the number of spikelets per plant, the number of kernels per spikelet or the grain weight, resulting in the wheat yield decline (Arduini, Baldanzi & Pampana, 2019; Hossain, Araki & Takahashi, 2011). Therefore, we suggest that comprehensive consideration of the changes of each index after waterlogging and drainage, combined with the final yield and yield components, can better evaluate the response of different peanut varieties to waterlogging, which provides a foundation for waterlogging tolerance breeding and high-yield cultivation.

At present, the comprehensive evaluation of multiple indicators of peanut to waterlogging needs to be further studied. In this experiment, the soil and plant analysis development (SPAD) value, chlorophyll fluorescence parameters, and enzyme activities of 15 peanut varieties after 7 days of waterlogging and 7 days of drainage were measured, combined with the yield and yield components of peanut at the harvest stage. Waterlogging tolerance and recovery capability of different peanut varieties were classified by principal component analysis, membership function analysis, and cluster analysis. The main screening indexes to identify the waterlogging tolerance and recovery capability of plants were determined. The relationships between waterlogging tolerance, recovery capability, yield and yield components of different peanut varieties were discussed, exploring the response mechanism of different peanut varieties to waterlogging stress and establishing a reliable comprehensive evaluation method for identifying peanut waterlogging tolerance. This study lays a foundation for selecting waterlogging tolerant varieties and the evaluation of waterlogging tolerance varieties in the peanut production areas of China.

Materials & Methods

Plant materials

Fifteen peanut varieties were selected as plant materials in this experiment, and the details of the materials were given in Table 1. The field experiment was conducted at the Zengcheng Teaching and Research Farm (23°24′N, 113°64′E) of South China Agricultural University (SCAU), which is located in Zengcheng District, Guangzhou City, Guangdong Province, China.

Table 1 Peanut varieties used in this study.

Varieties	Abbreviations	Supplier references	
Yueyou 13	YY 13	Guangdong Academy of Agricultural Sciences Guangzhou, China	
Hanghua 2	HH 2	Guangdong Academy of Agricultural Sciences, Guangzhou, China	
Yueyou 45	YY 45	Guangdong Academy of Agricultural Sciences, Guangzhou, China	
Heyou 4	HY 4	Guangxi Academy of Agricultural Sciences, Nanning, China	
Heyou 10	HY 10	Guangxi Academy of Agricultural Sciences, Nanning, China	
Yuhua 65	YH 65	Henan Academy of Agricultural Sciences, Zhengzhhou, China	
Yuhua 22	YH 22	Henan Academy of Agricultural Sciences, Zhengzhhou, China	
Dongbeiwang	DBW	Jilin Academy of Agricultural Sciences, Changchun, China	
Fuhua 1	FH 1	Jilin Academy of Agricultural Sciences, Changchun, China	
Jihua 16	JH 16	Liaoning Academy of Agricultural Sciences, Shenyang, China	
Kainong 1715	KN 1715	Shandong Academy of Agricultural Sciences, Jinan, China	
Huayu 39	HY 39	Shandong Academy of Agricultural Sciences, Jinan, China	
Jinhua 7	JH 7	Shanxi Academy of Agricultural Sciences, Taiyuan, China	
Yushehuasheng	YSHS	Shanxi Academy of Agricultural Sciences, Taiyuan, China	
Sililanka	SLLK	Sri Lanka	

Waterlogging treatment

Seeds of per variety were planted in three replicate plots. Waterlogging treatment was applied to peanut plants during the pod filling stage, and the water level was kept at 2 cm higher than the soil surface during the waterlogging process. And the blank control group of each variety was also set, During the growth stage, the CK groups of all varieties were irrigated normally and kept the soil moisture at 75–80% of saturated water-holding capacity. The water in the fields was drained after waterlogging for 7 days. Other management practices followed conventional cultivation methods.

Determination of SPAD value and chlorophyll fluorescence parameters

After 7 days of waterlogging and 7 days of drainage, the SPAD value in the functional leaves of the main stem (from the top of the main stem to the base of the stem, the third open leaf) was measured using a chlorophyll meter (SPAD-502; Konica Minolta Sensing, Inc., Osaka, Japan). Chlorophyll fluorescence parameters of the same leaf were analyzed using a hand-held device MultispeQ Beta (Kuhlgert et al., 2016). And PS actual quantum yield (ΦPS II), maximum photochemical efficiency (Fv/Fm), non-photochemical quenching (NPQ), and photochemical quenching (qL) of leaves were obtained. Three functional leaves were measured in each plot. The data of one plot was a replicate, and each treatment contained three replicates.

Determination of the activities of SOD and CAT

SOD activity was measured by monitoring the inhibition of nitro blue tetrazolium (NBT) reduction with a spectrophotometer at 560 nm; 100 µL of crude enzyme solution was added to 2.9 mL reaction solution, which was composed of 50 mM sodium phosphate buffer (pH 7.8), 60 µM riboflavin, 195 mM methionine, 3 µM ethylenediamine tetraacetic acid (EDTA), and 1.125 mM NBT; three mL reaction solution was used as the control. The mixture was placed under a 4000 lx fluorescent lamp for 60 min for chromogenic reaction, and then turned into the darkness to stop the reaction. One unit of SOD activity was defined as the amount of enzyme that inhibits the NBT reduction by 50%.

For estimating the CAT activity, 100 µL of enzyme extract was added to a mixture containing 5.9 mM of H2O2 and 50 mM of buffer. Recording the absorbance at 240 nm at per minute interval for 3 min. A unit of CAT activity was defined as the changes in absorbance at 240 nm per minute.

Determination of MDA content

MDA content was calculated by the thiobarbituric acid (TBA) method described by a previous report with modifications (Wassie et al., 2019). Fresh leaf samples (0.5 g) were put into liquid nitrogen and ground to powder, then were homogenized in 2 mL of 10% (v/v) TCA solution, followed by centrifugation at 3,000× g for 10 min and collection of the supernatant. The 2 mL obtained supernatant was added to an equal volume of reaction mixture, which contained 20% (v/v) trichloroacetic acid and 0.5% (v/v) thiobarbituric acid. The mixture was then heated at 100 °C water bath for 20 min and then was stopped by an ice bath, followed by centrifugation at 4,000 g for 10 min at 20 °C. To determine the MDA level, the supernatant absorbance was measured spectrophotometrically at 450, 532, and 600 nm.

Determination of relative electrolyte linkage (EL)

Electrolyte leakage was measured following the method described by the method of predecessors (Huang et al., 2017). 1.0 g fresh leaf samples were washed with deionized water for three times and then transferred to a 50-mL centrifuge tube containing deionized water. The test tube was cultured in a conical shaker at room temperature for 12 h, and a conductivity meter (Jenco-3173; Jenco Instruments, Inc., San Diego, CA, USA) was used to measure the initial electrical conductivity (EL1). Then the leaves were sterilized at 100 °C for 30 min, then cooled at room temperature, and the secondary conductivity (EL2) of the leaves was measured. The relative electrolyte linkage (REL) is calculated by the formula: (1) REL%=EL1/EL2×100.

Determination of yield and yield components

Plant samples were harvested on 15 December 2020 for the determination of the yield and yield components. The representative plant samples from each plot were obtained at the physiological maturity stage to determine the yield and yield components, including the pod yields per hectare (Y), the number of total pods per plant (TP), the hundred pods weight (HPW), and the hundred kernels weight (HKW). The proportion of reduction in yield and yield components is also calculated, including the proportion of reduction in yield (RY), the proportion of reduction in HPW (RHPW), the proportion of reduction in HKW (RHKW), and the proportion of reduction in TP (RTP). The data of one plot was a replicate, and each treatment contained three replicates.

Evaluation of waterlogging tolerance

The waterlogging tolerance coefficient (WTC) of all indicators was calculated using the following equation: (2) WTC=WK/CK×100

where CK is the mean value of an indicator under the control treatment and WK is the mean value of an indicator under waterlogging treatment.

Membership function values of various indicators of different varieties: (3) uXj=Xj−Xmin/Xmax−Xminj=1,2,…,n.

where µ(Xj) represents the subordinate function value of the j-th comprehensive indicator, Xj represents the j-th comprehensive indicator value, Xmax represents the maximum value of the j-th comprehensive indicator, and Xmin represents the minimum value of the j-th comprehensive indicator.

The index weight calculation formula of each comprehensive indicator is as follows: (4) wj=pj/∑j=1npjj=1,2,…,n

where wj is the index weight of the j-th comprehensive indicator among all comprehensive indicators and pj is the contribution rate of the j-th comprehensive indicator.

The comprehensive evaluation value of waterlogging tolerance of different peanut varieties: (5) D= ∑jnuXj×wjj=1,2,…,n

where D indicates the comprehensive evaluation value of waterlogging tolerance of peanut varieties. The higher the D value, the stronger the tolerance to waterlogging stress of peanut varieties; the lower the D value, the weaker the tolerance to waterlogging stress of peanut varieties.

Statistically analysis

Experimental data was statistically analyzed using Microsoft Excel 2010 and SPSS 22.0 (SPSS., Chicago, IL, USA), and the image was generated using Origin 2017. All data are means of three replicates (n = 3). Comparisons among multiple groups were performed using Fisher’s protected least significant difference (LSD) test. Probability values p < 0.05 were considered statistically significant. SPSS 22.0 software was used for variance analysis, principal component analysis, and cluster analysis of data. And the D value of different varieties was analyzed by cluster analysis at an Euclidean distance of 5.

Results

Response of different indexes to waterlogging and correlation analysis among them

Figures S1–S11 displayed the response of leaf indexes of 15 peanut varieties to 7 days of waterlogging and 7 days of drainage. Waterlogging significantly affected the photosynthetic characteristics and antioxidant capacity of peanut leaves. However, there were great differences in the response of different varieties to waterlogging. As shown in Table 2, after waterlogging for 7 days, the WTC of REL, MDA content, SOD activity, and CAT activity, NPQ and qL of leaves of some peanut varieties (HH 2, YH 65, YH 22) increased, and the WTC of SPAD value, ΦPS II, Fv/Fm of peanut varieties (KN 1715, DBW and SLLK) decreased. After 7 days of drainage, the WTC of REL, MDA content, SOD activity and CAT activity of leaves of peanut variety YY 45 decreased compared with that of 7 days of waterlogging, but these indicators of YSHS still increased. And after drainage, the WTC of SOD activity and CAT activity of leaves of peanut varieties (KN 1715, JH 7) decreased compared with that of after waterlogging for 7 days, but these indicators of (YY 13, HH 2, DBW, HY 10, and HJ 16) still increased. Besides, after 7 days of drainage, ΦPS II, Fv/Fm, NPQ and qL increased or decreased to different degrees in comparison with those after 7 days of waterlogging. And the WTC of HPW, HCW, Y, and TP of most varieties (YYHS, HH 2, KN 1715, HY 4, DBW, YH 65, YH 22, JH 7, HY 10, FH 1, JH 16, and HY 39) decreased, indicating that waterlogging had a negative impact on yield and yield components.

Table 2 Waterlogging tolerance coefficient of physiological indicators of leaves of peanut varieties after waterlogging for 7 days and drainage for 7 days, and yield and yield components at the harvest stage.

Varieties	YYHS	YY 13	HH 2	KN 1715	HY 4	DBW	YH 65	YH 22	JH 7	HY 10	FH 1	SLLK	JH 16	HY 39	YY 45	
REL(W)	0.785	1.808	1.320	1.039	1.086	1.092	1.098	1.718	0.585	1.002	0.940	1.980	1.133	0.968	1.260	
MDA(W)	0.888	1.674	1.149	1.040	1.262	0.563	1.402	1.468	1.224	0.870	1.953	1.178	0.876	0.825	1.191	
SOD(W)	1.727	0.958	1.443	1.745	2.264	0.555	1.679	3.125	1.241	1.099	0.993	0.962	1.693	1.148	1.285	
CAT(W)	0.931	1.669	1.558	0.975	0.955	0.683	1.917	1.243	0.893	2.466	8.216	1.862	1.759	0.869	9.530	
ΦPS II(W)	0.992	1.102	0.991	0.962	1.256	0.987	1.028	1.006	1.033	1.002	1.015	0.966	0.996	1.104	1.250	
Fv/Fm(W)	0.987	1.002	0.972	0.999	0.998	0.971	0.984	0.964	0.978	0.984	0.959	0.951	1.001	1.037	1.024	
NPQ(W)	1.684	0.954	1.954	1.214	0.368	1.850	1.405	2.847	1.276	1.444	2.215	1.925	1.201	0.243	0.100	
qL(W)	1.045	1.333	1.122	0.916	1.114	1.133	1.162	1.221	1.149	1.087	1.252	1.053	1.027	0.885	1.071	
SPAD(W)	0.895	0.987	0.919	0.971	1.038	0.787	1.128	0.925	1.094	0.892	0.912	0.702	0.955	1.116	1.027	
REL(D)	0.966	1.134	1.064	0.951	1.258	0.619	1.118	0.895	0.943	0.713	0.950	1.290	0.981	0.964	1.095	
MDA(D)	1.480	1.326	2.132	1.126	1.059	0.742	1.046	1.808	1.260	0.992	1.369	1.057	0.886	1.080	0.833	
SOD(D)	2.983	1.36	1.734	1.282	1.376	3.062	2.451	1.042	0.531	2.203	1.883	3.562	3.086	0.756	1.227	
CAT(D)	2.700	3.059	3.648	0.395	1.001	1.838	1.454	2.065	0.851	4.067	3.500	0.725	3.141	3.334	3.534	
ΦPS II(D)	1.115	1.133	1.033	0.936	1.148	1.074	0.972	0.917	0.967	0.913	1.168	0.929	1.025	1.012	0.973	
Fv/Fm(D)	1.109	1.007	0.979	0.986	1.08	0.998	0.998	0.985	0.996	0.951	0.972	0.941	0.991	1.044	1.036	
NPQ(D)	0.588	0.993	1.418	1.224	0.342	0.786	1.044	1.243	1.119	2.125	1.419	2.234	0.958	0.694	0.746	
qL(D)	0.918	1.401	1.226	0.876	1.018	1.321	0.928	0.831	0.901	0.960	0.874	1.033	1.140	0.804	0.886	
SPAD(D)	1.135	0.776	0.925	0.920	0.995	0.670	1.012	0.916	0.891	0.858	1.133	0.704	0.979	0.983	1.164	
HPW	0.858	0.957	0.771	0.591	0.783	0.841	0.652	0.950	0.554	0.617	0.977	0.985	0.697	0.764	0.582	
HKW	0.835	0.915	0.824	0.701	0.736	0.800	0.798	0.872	0.490	0.536	0.797	1.063	0.754	0.859	0.519	
TP	0.850	1.017	0.651	0.359	0.558	0.449	0.666	0.483	0.776	0.610	0.522	0.401	0.702	0.391	1.055	
Y	0.708	0.973	0.501	0.213	0.437	0.378	0.434	0.459	0.430	0.376	0.510	0.395	0.505	0.299	0.614	
Notes.

REL relative electrolyte linkage

MDA malondialdehyde content

SOD superoxide dismutase activity

CAT catalase activity

ΦPS II PSIIactual quantum yield

Fv/Fm maximum photochemical efficiency

NPQ non-photochemical quenching

qL photochemical quenching

SPAD the soil and plant analysis development

HPW hundred pods weight

HKW hundred kernels weight

Y yield per hectare

TP the number of total pods per plant

W waterlogging for 7 days

D drainage for 7 days

Table 3 Correlation coefficient matrix among all indicators of peanut varieties.

	REL(W)	MDA(W)	SOD(W)	CAT(W)	ΦPS II(W)	Fv/Fm(W)	NPQ(W)	qL(W)	SPAD(W)	REL(D)	MDA(D)	SOD(D)	CAT(D)	ΦPS II(D)	Fv/Fm(D)	NPQ(D)	qL(D)	SPAD(D)	HPW	HKW	TP	Y	
REL(W)	1																						
MDA(W)	0.293	1																					
SOD(W)	0.110	0.181	1																				
CAT(W)	−0.007	0.446	−0.216	1																			
ΦPS II(W)	−0.017	0.174	0.114	0.377	1																		
Fv/Fm(W)	−0.244	−0.280	−0.021	0.082	0.605*	1																	
NPQ(W)	0.245	0.218	0.199	−0.139	−0.754**	−0.878**	1																
qL(W)	0.317	0.699**	0.013	0.207	0.062	−0.473	0.402	1															
SPAD(W)	−0.445	0.186	0.248	0.030	0.506	0.646**	−0.585*	−0.073	1														
REL(D)	0.422	0.449	0.190	0.100	0.422	0.090	−0.311	0.053	0.179	1													
MDA(D)	0.181	0.381	0.409	−0.149	−0.265	−0.362	0.534*	0.315	−0.037	0.118	1												
SOD(D)	0.191	−0.320	−0.274	−0.096	−0.464	−0.428	0.341	−0.056	−0.689**	−0.036	−0.268	1											
CAT(D)	−0.043	0.020	−0.229	0.443	0.115	0.269	−0.067	0.129	0.020	−0.273	0.173	−0.017	1										
ΦPS II(D)	−0.173	0.280	−0.188	0.123	0.296	0.049	−0.122	0.370	0.026	0.138	0.076	0.071	0.212	1									
Fv/Fm(D)	−0.375	−0.181	0.254	−0.089	0.547*	0.534*	−0.492	−0.193	0.396	0.178	−0.039	−0.164	0.008	0.508	1								
NPQ(D)	0.373	0.132	−0.236	0.069	−0.564*	−0.581*	0.501	0.081	−0.537*	−0.059	0.138	0.283	0.048	-.531*	-.857**	1							
qL(D)	0.374	−0.100	−0.400	−0.251	−0.081	−0.140	0.053	0.405	−0.354	−0.009	0.003	0.352	0.120	0.385	−0.140	−0.021	1						
SPAD(D)	−0.463	0.263	0.323	0.553*	0.381	0.363	−0.259	−0.131	0.494	0.181	0.122	−0.245	0.337	0.242	0.518*	−0.415	−0.588*	1					
HPW	0.582*	0.386	0.011	−0.032	−0.175	−0.469	0.509	0.454	−0.547*	0.174	0.341	0.303	0.068	0.451	−0.050	0.126	0.286	−0.236	1				
HKW	0.645**	0.162	0.033	−0.306	−0.303	−0.311	0.377	0.110	−0.438	0.352	0.276	0.417	−0.128	0.224	−0.067	0.125	0.292	−0.329	0.829**	1			
TP	−0.019	0.215	−0.098	0.360	0.429	0.313	−0.341	0.397	0.295	0.209	0.022	−0.120	0.381	0.238	0.345	−0.295	0.249	0.338	−0.190	−0.343	1		
Y	0.298	0.410	−0.085	0.214	0.262	0.082	−0.060	0.621*	0.012	0.273	0.227	0.034	0.388	0.518*	0.303	−0.231	0.468	0.138	0.370	0.166	0.820**	1	
Notes.

REL relative electrolyte linkage

MDA malondialdehyde content

SOD superoxide dismutase activity

CAT catalase activity

ΦPS II PSIIactual quantum yield

Fv/Fm maximum photochemical efficiency

NPQ non-photochemical quenching

qL photochemical quenching

SPAD the soil and plant analysis development

HPW hundred pods weight

HKW hundred kernels weight

Y yield per hectare

TP the number of total pods per plant

W waterlogging for 7 days

D drainage for 7 days

* A significant difference (p < 0.05).

** A significant difference (p < 0.01).

Besides, from the correlation coefficient matrix (Table 3), there was a positive correlation between Fv/Fm and SPAD value of different peanut varieties after 7 days of waterlogging and 7 days of drainage (0.646 and 0.518, respectively). Meanwhile, HPW and HKW showed an extremely significant positive correlation (0.829), and Y was significantly positively correlated with TP (0.820). However, there may be information overlap between different indicators, and every single indicator plays a different role in the waterlogging tolerance of peanut. Therefore, it is difficult to accurately evaluate the waterlogging tolerance of different peanut varieties by using these indicators directly. To make up the deficiency of a single indicator evaluation of waterlogging tolerance, the principal component analysis method was used.

Comprehensive evaluation of waterlogging tolerance of different peanut varieties

Principal component analysis, membership function analysis and comprehensive evaluation

To eliminate the factors with small influence and large interference and improve the accuracy of measurement data analysis, the above single indicator was converted into a fewer and more effective indicator. Therefore, Principal component analysis was conducted based on the WTC of 9 indicators of 15 peanut cultivars after waterlogging for 7 days. As shown in Table 4, three principal components were selected. And the contribution rates of the comprehensive indexes CI1 to CI 3 were 36.759%, 24.092%, and 11.244%, respectively, and the eigenvalue was 3.308, 2.168, and 1.299, respectively. The cumulative contributions to the total variation of the population from first to third of principal component reached over 75.289%, which represented that the tested materials could be used for further analysis.

Table 4 Eigenvalue and contribution of each comprehensive index after waterlogging for 7 days and loading matrix of each component.

Items	Traits	Principal component	
		1	2	3	
Eigenvalue	3.308	2.168	1.299	
Contributive ratio (%)	36.759	24.092	14.438	
Cumulative contribution (%)	36.759	60.851	75.289	
Loading matrix of each component	REL(W)	−0.448	0.293	−0.009	
MDA(W)	−0.273	0.880	0.104	
SOD(W)	−0.032	0.163	0.893	
CAT(W)	0.122	0.622	−0.546	
ΦPS II(W)	0.699	0.546	−0.022	
Fv/Fm(W)	0.932	−0.011	0.009	
NPQ(W)	−0.938	−0.067	0.173	
qL(W)	−0.494	0.710	0.012	
SPAD(W)	0.732	0.297	0.403	
Notes.

REL relative electrolyte linkage

MDA malondialdehyde content

SOD superoxide dismutase activity

CAT catalase activity

ΦPS II PSIIactual quantum yield

Fv/Fm maximum photochemical efficiency

NPQ non-photochemical quenching

qL photochemical quenching

SPAD the soil and plant analysis development

W waterlogging for 7 days

According to Eq. (3), the membership function values of all the comprehensive indexes of different peanut varieties were obtained (Table 5). From the perspective of a single comprehensive index, such as CI1, the u (X1) of HY 39 is the largest, which is 1.0000, indicating that HY 39 has the strongest tolerance to waterlogging in CI1. On the contrary, the u (X1) of SLLK is the smallest, which is 0.0000, indicating that SLLK has the weakest tolerance to waterlogging in this comprehensive index. Since the contribution rates of comprehensive indexes CI1∼CI 3 are different, Eq. (4) is used to calculate the weights of each comprehensive index, which are 0.4882, 0.3200, and 0.1918, respectively.

Table 5 Value of each comprehensive indicators [CIx], subordinate function values (Xj), comprehensive evaluation value (D) and order after waterlogging for 7 days.

Varieties	CI 1	CI 2	CI 3	u ( X 1 )	u ( X 2 )	u ( X 3 )	D1 value	Order	
YYHS	−0.1830	−1.4448	0.4030	0.4179	0.1117	0.4779	0.3314	11	
YY 13	−0.1998	2.2024	−0.3663	0.4151	0.9584	0.2910	0.5652	4	
HH 2	−1.2226	−0.3380	0.1238	0.2493	0.3687	0.4101	0.3183	12	
KN 1715	0.6199	−1.5418	0.5873	0.5480	0.0892	0.5227	0.3963	10	
HY 4	2.0087	1.0857	1.2827	0.7731	0.6992	0.6917	0.7339	2	
DBW	−1.2791	−1.9260	−1.3938	0.2402	0.0000	0.0413	0.1252	14	
YH 65	0.2387	0.7659	0.9428	0.4862	0.6249	0.6091	0.5542	5	
YH 22	−2.5916	1.0813	2.5514	0.0274	0.6982	1.0000	0.4286	7	
JH 7	0.5206	−0.1202	0.4115	0.5319	0.4192	0.4800	0.4859	6	
HY 10	−0.2201	−1.0123	−0.7174	0.4118	0.2121	0.2057	0.3084	13	
FH 1	−1.8226	2.3815	−1.3738	0.1521	1.0000	0.0462	0.4031	9	
SLLK	−2.7607	−0.7288	−1.2058	0.0000	0.2779	0.0870	0.1056	15	
JH 16	0.5218	−1.0908	0.2897	0.5321	0.1939	0.4504	0.4082	8	
HY 39	3.4080	−1.3344	0.0287	1.0000	0.1374	0.3870	0.6064	3	
YY 45	2.9616	2.0203	−1.5638	0.9276	0.9161	0.0000	0.7461	1	
Index weight				0.4882	0.3200	0.1918			
Notes.

“CI” values were determined by principal component analysis (PCA) and the scores of the comprehensive indicators.

Cluster analysis of different peanut varieties based on D1 value after 7 days of waterlogging

The higher the D1 value, the stronger the tolerance to waterlogging stress of peanut varieties; the lower the D1 value, the weaker the tolerance to waterlogging stress of peanut varieties. As indicated in Table 5, the D1 value of YY 45 ranked first, followed by HY 4, and the D1 value of SLLK ranked last, indicating that YY 45 has the highest waterlogging tolerance while SLLK has the lowest. Therefore, we used the hierarchical cluster analysis to cluster D1 values. Figure 1 displayed that 15 peanut varieties could be divided into three categories: waterlogging-tolerant varieties (HY 4, YY 45, YY 13, YH 65, and HY 39), intermediate varieties (FH 1, JH 16, KN 1715, YH 22, JH 7, HH 2, HY 10, and YSHS) and waterlogging-sensitive varieties (DBW and SLLK).

Figure 1 Hierarchical cluster analysis.

(A) Hierarchical cluster analysis based on the D1 value after waterlogging for 7 days to evaluate the waterlogging tolerance of 15 peanut varieties; (B) hierarchical cluster analysis based on the D2 value after drainage for 7 days to evaluate the recovery capability of 15 peanut varieties.

Comprehensive evaluation of recovery capability of different peanut varieties

Principal component analysis, membership function analysis and comprehensive evaluation

A principal component analysis was also conducted based on the WTC of 9 indicators of 15 peanut cultivars after drainage for 7 days, and four principal components were selected (Table 6). And the contribution rates of the comprehensive indexes CI 1 to CI 4 were 30.493%, 19.614%, 15.385%, and 13.022%, respectively, and the eigenvalue was 2.744, 1.765, 1.385, and 1.172, respectively. The cumulative contributions to the total variation of the population from first to fourth of principal component reached over 78.514%, which indicated that the tested materials could be used for further analysis. And the membership function values of all the comprehensive indexes of different peanut varieties were obtained according to Eq. (3) (Table 7). Equation (4) is used to calculate the weights of each comprehensive index, which are 0.3884, 0.2498, 0.1960, and 0.1659, respectively.

Table 6 Eigenvalue and contribution of each comprehensive index after drainage for 7 days and loading matrix of each component.

Items	Traits	Principal component	
		1	2	3	4	
Eigenvalue	2.744	1.765	1.385	1.172	
Contributive ratio (%)	30.493	19.614	15.385	13.022	
Cumulative contribution (%)	30.493	50.106	65.492	78.514	
Loading matrix of each component	REL(D)	0.237	−0.017	−0.414	0.694	
MDA(D)	0.079	−0.234	0.51	0.702	
SOD(D)	−0.371	0.544	−0.074	−0.186	
CAT(D)	0.155	0.055	0.879	−0.202	
ΦPS II(D)	0.599	0.630	0.195	0.156	
Fv/Fm(D)	0.897	0.160	−0.176	−0.089	
NPQ(D)	−0.857	−0.254	0.201	0.155	
qL(D)	−0.248	0.869	0.165	0.229	
SPAD(D)	0.749	−0.411	0.198	−0.109	
Notes.

REL relative electrolyte linkage

MDA malondialdehyde content

SOD superoxide dismutase activity

CAT catalase activity

ΦPS II PSIIactual quantum yield

Fv/Fm maximum photochemical efficiency

NPQ non-photochemical quenching

qL photochemical quenching

SPAD the soil and plant analysis development

D drainage for 7 days

Cluster analysis of different peanut varieties based on D2 value after 7 days of drainage

The higher the D2 value, the higher the recovery capability of peanut varieties; the lower the D2 value, the lower the recovery capability of peanut varieties. Therefore, we used the hierarchical cluster analysis to cluster D2 values. The D2 value of YY 13 ranked first, followed by YSHS, and the D2 value of SLLK ranked last, indicating that YY 13 has the highest recovery capability while SLLK has the lowest (Table 7). Figure 1 revealed that 15 peanut varieties could be divided into three categories: varieties with high recovery capability (YSHS, YY 13, HH 2, HY 4, and FH 1), varieties with intermediate recovery capability (YH 65, YH 22, JH 7, JH 16, HY 39, YY 45, and DBW), varieties with low recovery capability (KN 1715, HY 10, and SLLK).

The indicators used for identifying waterlogging tolerance and recovery capability and their relationship with yield and yield components

Correlation analysis was carried out between the D1 value and the WTC of each physiological index after 7 days of waterlogging (Table 8). The D1 value strongly and positively correlated with the ΦPSII(W), SPAD(W), Fv/Fm(W), and the correlation coefficient was 0.845, 0.845, 0.708, respectively. Besides, there was a significant negative correlation between NPQ(W) and D1 value, and the WTC was −0.707. Therefore, the WTC of ΦPS II(W), NPQ(R), Fv/Fm(W), and SPAD values measured after 7 days of waterlogging can be used as important indexes for the identification of waterlogging tolerance.

Table 7 Value of each comprehensive indicators [CIx], subordinate function values (Xj), comprehensive evaluation value (D) and order after drainage for 7 days.

Varieties	CI 1	CI 2	CI 3	CI 4	u(X1)	u(X2)	u(X3)	u(X4)	D2 value	Order	
YYHS	2.6275	0.6039	0.3227	−0.3109	1.0000	0.5311	0.5354	0.3765	0.6884	2	
YY 13	0.1073	2.1252	0.6002	1.3939	0.5728	0.8538	0.6085	0.8193	0.6909	1	
HH 2	−0.4490	0.2235	2.0866	2.0893	0.4784	0.4504	1.0000	1.0000	0.6601	3	
KN 1715	−0.6668	−1.2722	−1.3463	−0.1249	0.4416	0.1331	0.0959	0.4248	0.2940	13	
HY 4	2.5849	0.8691	−1.6734	0.8036	0.9928	0.5874	0.0097	0.6660	0.6447	4	
DBW	−1.2761	2.8141	−0.2517	−1.7602	0.3383	1.0000	0.3842	0.0000	0.4565	9	
YH 65	−0.0271	−0.4158	−1.0897	−0.0510	0.5500	0.3148	0.1635	0.4440	0.3979	10	
YH 22	−0.5213	−1.8997	0.5226	0.5798	0.4663	0.0000	0.5881	0.6079	0.3971	11	
JH 7	−0.2020	−1.2573	−0.8692	0.2047	0.5204	0.1363	0.2215	0.5104	0.3642	12	
HY 10	−2.4813	−0.7824	1.4902	−1.5247	0.1340	0.2370	0.8429	0.0612	0.2866	14	
FH 1	0.7142	−0.3302	1.5690	0.0256	0.6757	0.3329	0.8637	0.4639	0.5918	5	
SLLK	−3.2722	0.2331	−1.7103	1.1852	0.0000	0.4525	0.0000	0.7651	0.2399	15	
JH 16	−0.2353	1.1281	0.1943	−0.8512	0.5148	0.6423	0.5016	0.2361	0.4978	7	
HY 39	1.4682	−1.0098	0.1862	−0.7596	0.8035	0.1888	0.4995	0.2599	0.5002	6	
YY 45	1.6298	−1.0297	−0.0311	−0.8998	0.8309	0.1846	0.4423	0.2235	0.4925	8	
Index weight				0.3884	0.2498	0.1960	0.1659			
Notes.

“CI” values were determined by principal component analysis (PCA) and the scores of the comprehensive indicators.

Table 8 Correlation analysis between D1 value and physiological indexes measured after 7 days of waterlogging.

	REL(W)	MDA(W)	SOD(W)	CAT(W)	ΦPS II(W)	Fv/Fm(W)	NPQ(W)	qL(W)	SPAD(W)	
D1	−0.175	0.332	0.322	0.300	0.845**	0.708**	−0.707**	0.038	0.845**	
Notes.

REL relative electrolyte linkage

MDA malondialdehyde content

SOD superoxide dismutase activity

CAT catalase activity

ΦPS II PSII actual quantum yield

Fv/Fm maximum photochemical efficiency

NPQ non-photochemical quenching

qL photochemical quenching

SPAD the soil and plant analysis development

W waterlogging for 7 days

* A significant difference (p < 0.05).

** A significant difference (p < 0.01).

The correlation between D2 value and the WTC of each physiological index after 7 days of drainage was also analyzed (Table 9). A significant positive correlation of D2 value was observed with ΦPSII(D) (r = 0.855). And there was a positive relation between D2 value and Fv/Fm(D)(r = 0.625). However, there was a significant negative correlation between NPQ(D) and D2 value, and the correlation coefficient was −0.609. Therefore, the WTC of ΦPS II(D), Fv/Fm(D) and NPQ(D) measured after 7 days of drainage can be used as important indexes for the identification of recovery capability.

Table 9 Correlation analysis between D2 value and physiological indexes measured after 7 days of drainage.

	REL(D)	MDA(D)	SOD(D)	CAT(D)	ΦPS II(D)	Fv/Fm(D)	NPQ(D)	qL(D)	SPAD(D)	
D2	0.212	0.370	−0.102	0.429	0.855**	0.625*	−0.609*	0.364	0.396	
Notes.

REL relative electrolyte linkage

MDA malondialdehyde content

SOD superoxide dismutase activity

CAT catalase activity

ΦPSII PSIIactual quantum yield

Fv/Fm maximum photochemical efficiency

NPQ non-photochemical quenching

qL photochemical quenching

SPAD the soil and plant analysis development

D drainage for 7 days

* A significant difference (p < 0.05).

** A significant difference (p < 0.01).

To determine the relationship between D1 value, D2 value, and the proportion of reduction in yield and yield components, a linear model among them was fitted. As shown in Fig. 2, the R2 between the RTP and RY is the highest (0.6732), followed by the R2 between the D2 value and the RY (0.4458). In conclusion, both the D2 value and the RTP have significant effects on the final yield. The lower the RTP and the higher the D2 value, the less the yield loss under waterlogging. However, there was no significant correlation between the D1 value and RY. In addition, no significant correlation was found between the D1 value and D2 value, suggesting that the waterlogging tolerance of the variety did not affect the recovery capability.

Figure 2 The linear fit between the D1 value, D2 value, and the proportion of reduction in yield and yield components.

(A) Between the D1 value and D2 value; (B) Between the D1 value and the proportion of reduction in yield (RY); (C) Between the value of.

Yield and yield components of peanut varieties with different waterlogging recovery capability

Peanut with different recovery capacities had different yield decline under waterlogging stress (Fig. 3). For example, YY 13 is a cultivar with high recovery capability and YY 45 is a cultivar with intermediate recovery capability. And RTP of the two cultivars were −1.67% and −3.51%. However, the RY of YY 13 was only 2.71%, while that of YY 45 was 39.76%. Besides, HH 2 belongs to the variety with high recovery capability, and YH 65 belongs to the variety with intermediate recovery capability. It is found that the TP of the two varieties decreased by 34.89% and 33.40% respectively under waterlogging, which did not reach a significant difference, but the yield of HH 2 decreased by 49.86%, while the yield of YH 65 decreased by 56.61%. It showed that the recovery capability of peanut affected the final pod yield.

Figure 3 The proportion of reduction in yield and yield components.

(A) The proportion of reduction in yield (RY); (B) the proportion of reduction in HPW (RHPW); (C) the proportion of reduction in HKW (RHKW); (D) the proportion of reduction in TP (RTP).

And among peanut varieties with similar recovery capability, the RY varied greatly among varieties. Among the cultivars with high recovery capability (YY 13, YSHS, HH 2, FH 1, HY 4), the D2 values of YY 13 and HY 4 were 0.6909 and 0.6447, respectively, but the RY of YY 13 and HY 4 was 2.71% and 56.32%, respectively. The RTP of YY 13 was 1.67% while that of HY 4 was 44.21%. In cultivars with intermediate recovery capability (YY 45, JH 7, JH 16, YH 65, YH 22, DBW, HY 39), the yield of YY 45 and HY 39 decreased by 38.62% and 70.08%, respectively, and the TP of YY 45 increased by 3.47% while that of HY 39 decreased by 60.91%. In the cultivars with low recovery capability, including HY 10, SLLK, and KN 1715, the RTP of HY 10, SLLK, and KN 1715 was 39.03%, 59.85% and 64.07%, respectively, and the RY of them was 62.35%, 60.46% and 78.74%, respectively. The results indicated that the yield of the peanut varieties with lower recovery capability decreased greatly, while the yield of the peanut varieties with the same recovery capability was affected by RTP.

Discussion

Peanut is abundant in oil and protein, which is not only an important oil crop, but also an important raw material for the food industry, medicine, and other industries, playing a critical role in economic development and agricultural development (Latif et al., 2013). Previous studies showed that waterlogging led to peanut rotten pods and finally lead to yield penalty (Bishnoi & Krishnamoorthy, 1992; Zeng et al., 2020). Due to the rising greenhouse gas levels, the likelihood of extreme rainfall may have doubled (Smethurst & Shabala, 2003). Therefore, the establishment of a comprehensive evaluation system to evaluate the waterlogging tolerance of peanut is of great significance to cultivation and breeding.

Different peanut varieties have different response mechanisms to waterlogging stress. Waterlogging stress would finally lead to cellular and membrane damage and loss of photosynthetic capacity of plant leaves (Zhang et al., 2015; Smethurst & Shabala, 2003; Irving et al., 2007; Singh et al., 2019). However, plants exposed to waterlogging are prone to damage at the cellular level and cause irreversible metabolic dysfunctions leading to cell death (Pereira et al., 2015). After drainage, in some plants, such as soybean, levels of photosynthetic pigments increased to pre-waterlogging level, and the enzyme activities of SOD and CAT returned to the normal level compared with the treatment without waterlogging (Seymen, 2021), but other study found that waterlogging stress release by soil drainage did not improve plant performance but, on the contrary, enhanced oxidative stress and even accelerated plant injury (Hossain et al., 2009). In this study, waterlogging led to the increase in REL, SOD activities, CAT activities, and MDA content, and the decrease of SPAD value and Fv/Fm in leaves of most peanut varieties, which demonstrated that the photosynthesis of most peanut cultivars was restricted after waterlogging stress, and reactive oxygen species and free radicals were accumulated in the cells of peanut. Besides, the indexes (physiological and biochemical parameters) of some varieties could return to the normal level after 7 days of drainage, indicating that for some peanut varieties, their physiological and biochemical parameters can be restored to normal levels after drainage. Clearly, waterlogging caused irreversible damage to the photosynthetic system and antioxidant system of these peanut varieties. But the tolerance of different peanut varieties to waterlogging could not be comprehensively evaluated based on the changes of these indexes.

Membership function analysis, principal component analysis, and cluster analysis have been widely used in plant tolerance evaluation (Liu et al., 2020; Aghaie et al., 2018; Seymen, 2021). The membership function analysis is often used for the comprehensive evaluation of crop waterlogging stress. However, there are some limitations on the comprehensive evaluation of waterlogging tolerance by the membership function analysis for the correlation between indicators (Pancardo et al., 2021). Besides, the principal component analysis can convert the original indexes into new comprehensive and independent indicators (Köksal, 2011). Therefore, on this basis, by weighing the value of each comprehensive index and the corresponding membership function value of varieties, the comprehensive evaluation value of each genotype can be obtained, so as to comprehensively evaluate the tolerance of different varieties (Li et al., 2018; Zhou et al., 2019). Therefore, to better clarify the response mechanism of peanut varieties to waterlogging, we evaluated the physiological indexes measured at 7 days of waterlogging and 7 days of drainage, respectively, and discussed the waterlogging tolerance and recovery capability of different peanut varieties. Results suggested that there were great differences in waterlogging tolerance and recovery capability among various peanut varieties. However, linear fit model revealed that there was no significant relationship between waterlogging tolerance and recovery capability, which indicated that varieties with high waterlogging tolerance did not necessarily have high recovery capability. For example, peanut varieties YY 45, YH 65, and Huayu 39, have high waterlogging tolerance, but their recovery capacity is not the strongest. In addition, we also found that the recovery capacity of varieties with weak waterlogging tolerance was also weak.

Prior studies indicated that recovery of physiological and biochemical parameters after drainage was closely related to the survival of plants under adverse conditions (Tian et al., 2021; Pompeiano et al., 2019). Linear fit model showed that there was no correlation between D1 value and the RY, and there was a significant negative correlation between the D2 value and the RY, indicating that the recovery capability after waterlogging was crucial to the final pod yield. While the study on cowpea suggested that the cultivars with higher drought tolerance were able to maintain higher photochemical activity and leaf gas exchange during water deficit than the sensitive cultivar did, which could alleviate the stress effects to the photosynthetic machinery and improve its recovery ability (Rivas et al., 2016). After waterlogging, the ability of legumes to retain green leaves was essential to succeed during recovery (Striker, Kotula & Colmer, 2019). Our study found that there was no significant difference in the RTP between YY 13 and YY 45, but there was a significant difference in the RY, which was speculated to be due to the different recovery capabilities of the two varieties. YY 13 had high recovery capability while YY 45 had intermediate recovery capability. The same pattern was also observed between cultivars HH 2 and YH 65. However, the recovery capacity of peanut variety was not the only factor that determined the final yield. The results showed that the RY was also related to the RTP, indicating that TP was affected under waterlogging. Besides, among the varieties with the same recovery capability, only the varieties with the lower RTP have less yield loss. For example, YY13 and YY45 were varieties with high recovery capability, but the decrease in TP of YY45 was higher than that of YY13. Therefore, the RTP was crucial to the final yield, which was in agreement with the linear fit analysis (Fig. 2). Therefore, the varieties with low recovery capability should not be considered when breeding, and the varieties with high recovery capability can be selected. In this study, we found that chlorophyll fluorescence parameters (ΦPS II, SPAD, Fv/Fm) were also important indicators to identify the recovery capability. This is similar to the results of previous researches (Wu et al., 2015; Smethurst, Garnett & Shabala, 2005). Therefore, it is suggested to select cultivars with smaller RTP under waterlogging. Also, the results of this study have a certain guiding significance for waterlogging tolerance breeding. We also think that varieties with higher waterlogging recovery ability should be selected in cultivation. It is indispensable to build water conservation projects and to drain water in time after waterlogging (Garrity & Pernito, 1996; Tian et al., 2019). At the same time, after waterlogging, taking appropriate measures to improve the recovery capability of peanut plants is necessary.

This study also showed that the variation trend of the same index of different peanut varieties was very different under the waterlogging stress and during the recovery process. At present, there are few studies on the physiological and biochemical response mechanism of peanut under waterlogging stress, and no comprehensive method has been provided for the comprehensive evaluation of its waterlogging tolerance. Many studies have shown that there is a correlation between waterlogging tolerance and morphological, physiological, and other indexes, so this study makes a comprehensive evaluation of several single indexes. The D value of each variety was obtained according to the weight of the corresponding comprehensive index, and the waterlogging tolerance and recovery capacity of each variety were obtained based on the D value. Meanwhile, for the first time, the relationship between waterlogging tolerance and recovery capacity and yield and yield components was discussed. This method can make up for the one-sidedness of single index analysis, the results obtained are more objective and scientific, and provide a reliable basis for practical application.

Conclusions

Different peanut varieties have different responses to waterlogging and drainage. It was found that some physiological and biochemical parameters of some peanut varieties could return to normal levels after drainage, but some varieties suffered irreversible damage from waterlogging. There was no significant relationship between waterlogging tolerance and recovery capability of different varieties, but there was a significant negative correlation between recovery capability and yield decline. Correlation analysis showed that fluorescence parameters (ΦPS II, Fv/Fm, qL) can be used as important screening indexes to identify the waterlogging recovery capability. It is a feasible method for breeding and cultivation to select cultivars with higher recovery capability and less reduction in pod numbers under waterlogging.

Supplemental Information

Supplemental Information 1 Raw data

Click here for additional data file.

Supplemental Information 2 Effects of waterlogging for 7 days and drainage for 7 days on relative electrolyte linkage (REL) of leaves of 15 peanut varieties

Data represents the mean ± standard error. Letters a and b represent statistically significant differences (p < 0.05 within a variety under control treatment (CK) and waterlogging treatment (WK) as determined by the least significant difference test.

Click here for additional data file.

Supplemental Information 3 Effects of waterlogging for 7 days and drainage for 7 days on malondialdehyde (MDA) content of leaves of 15 peanut varieties

Data represents the mean ± standard error. Letters a and b represent statistically significant differences (p < 0.05) within a variety under control treatment (CK) and waterlogging treatment (WK) as determined by the least significant difference test.

Click here for additional data file.

Supplemental Information 4 Effects of waterlogging for 7 days and drainage for 7 days on superoxide dismutase (SOD) activity of leaves of 15 peanut varieties

Data represents the mean ± standard error. Letters a and b represent statistically significant differences (p < 0.05) within a variety under control treatment (CK) and waterlogging treatment (WK) as determined by the least significant difference test.

Click here for additional data file.

Supplemental Information 5 Effects of waterlogging for 7 days and drainage for 7 days on catalase (CAT) activity of leaves of 15 peanut varieties

Data represents the mean ± standard error. Letters a and b represent statistically significant differences (p < 0.05) within a variety under control treatment (CK) and waterlogging treatment (WK) as determined by the least significant difference test.

Click here for additional data file.

Supplemental Information 6 Effects of waterlogging for 7 days and drainage for 7 days on PS actual quantum yield (Φ_PS) of leaves of 15 peanut varieties

Data represents the mean ± standard error. Letters a and b represent statistically significant differences (p < 0.05) within a variety under control treatment (CK) and waterlogging treatment (WK) as determined by the least significant difference test.

Click here for additional data file.

Supplemental Information 7 Effects of waterlogging for 7 days and drainage for 7 days on maximum photochemical efficiency (Fv/Fm) of leaves of 15 peanut varieties

Data represents the mean ± standard error. Letters a and b represent statistically significant differences (p < 0.05) within a variety under control treatment (CK) and waterlogging treatment (WK) as determined by the least significant difference test.

Click here for additional data file.

Supplemental Information 8 Effects of waterlogging for 7 days and drainage for 7 days on non-photochemical quenching (NPQ) of leaves of 15 peanut varieties

Data represents the mean ± standard error. Letters a and b represent statistically significant differences (p < 0.05) within a variety under control treatment (CK) and waterlogging treatment (WK) as determined by the least significant difference test.

Click here for additional data file.

Supplemental Information 9 Effects of waterlogging for 7 days and drainage for 7 days on photochemical quenching (qL) of leaves of 15 peanut varieties

Data represents the mean ± standard error. Letters a and b represent statistically significant differences (p < 0.05) within a variety under control treatment (CK) and waterlogging treatment (WK) as determined by the least significant difference test.

Click here for additional data file.

Supplemental Information 10 Effects of waterlogging for 7 days and drainage for 7 days on soil and plant analysis development (SPAD) value of leaves of 15 peanut varieties

Data represents the mean ± standard error. Letters a and b represent statistically significant differences (p < 0.05) within a variety under control treatment (CK) and waterlogging treatment (WK) as determined by the least significant difference test.

Click here for additional data file.

Supplemental Information 11 Effects of waterlogging for 7 days on yield and yield components at the harvest stage of 15 peanut varieties

Data represents the mean ± standard error. Letters a and b represent statistically significant differences (p < 0.05) within a variety under control treatment (CK) and waterlogging treatment (WK) as determined by the least significant difference test. HPW: hundred pods weight; HKW: hundred kernels weight; Y: yield per hectare; TP: the number of total pods per plant.

Click here for additional data file.

The authors are grateful to the reviewers and editors for their review and suggestions for this paper.

Additional Information and Declarations

Competing Interests

Author Contributions

Data Availability

The authors declare there are no competing interests.

Ruier Zeng and Jing Cao conceived and designed the experiments, performed the experiments, analyzed the data, prepared figures and/or tables, authored or reviewed drafts of the paper, and approved the final draft.

Xi Li conceived and designed the experiments, performed the experiments, analyzed the data, prepared figures and/or tables, and approved the final draft.

Xinyue Wang, Ying Wang, Suzhe Yao, Yu Gao, Jing Hu and Mingzhu Luo performed the experiments, analyzed the data, prepared figures and/or tables, and approved the final draft.

Lei Zhang conceived and designed the experiments, performed the experiments, analyzed the data, prepared figures and/or tables, and approved the final draft.

Tingting Chen conceived and designed the experiments, performed the experiments, analyzed the data, prepared figures and/or tables, authored or reviewed drafts of the paper, and approved the final draft.

The following information was supplied regarding data availability:

The raw measurements are available in the Supplementary Files.

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
