# Peer review of "Waterlogging tolerance and recovery capability screening in peanut: a comparative analysis of waterlogging effects on physiological traits and yield"

_PeerJ, doi:10.7717/peerj.12741_

## Round 0.1 · original submission · Major Revisions

Please consider the reviewers' comments and revise the manuscript accordingly. Also, provide a rebuttal letter.

·

Basic reporting

the English expression is not proper, I do highly recommend the help of English native speakers to help in polishing the manuscript.

Experimental design

Material and methods
Plant materials: the authors need to justify why the 15 lines were chosen, this will help the readers to understand the reason for this research work. Are they the best performing lines or potentially tolerant to waterlogging. This information needs to be made clear to the readers.
Waterlogging treatment: this section is highly mixed up, let the authors stick to waterlogging as a treatment and not mix with the data collection. The critical question is that can groundnuts survive in a waterlogged environment for a whole week, I doubt. So proper justification must be provided to the authors. Furthermore, SPAD and other antioxidant evaluation sections 142 to 148 must be removed. Moreover, it is better to state the dates in periods either after two weeks of planting or a month after planting as opposed to indicating the actual dates. This paper will be read beyond this century.
Determination of SPAD value and chlorophyll fluorescence parameters: one wonders what you meant by the term “ functional leaf” or “functional leaves” the nature of peanuts has nodes, so it would be proper to indicate the position of the measured leaf. What was the significance of taking the SPAD values in addition to the Chlorophyll fluorescence parameter??
Determination of the activities of SOD and CAT: SOD was evaluated as per the previous method though with modification, it would be proper to indicate the modifications, this will help in the repeatability of this particular research. Moreover, you have gone to write the whole process, so it doesn’t warrant the need to indicate the SOD measurements were as per the previous method.

Validity of the findings

Results
Before carrying out correlation, it would be good to carry out ANOVA of the various trait measured among the 15 accessions in this experiment. Once ANOVA is done, then the authors can do correlation analysis.
The authors need to avoid the term “As can be seen from Table 2” and consider replacing it with as shown or indicated
The results section is not well written, and this could be attributed to the very basic analysis; in the introduction, several methods were indicated such as “Membership function analysis, principal component analysis and cluster analysis” the cluster analysis helps in determining the hotspots while PCA applicable in data reduction. Can the authors help the readers to understand why these were mentioned and not shown under results?

Additional comments

Abstract
This work by Zeng et al “Waterlogging tolerance and recovery capability screening in peanut: a comparative analysis of waterlogging effects on physiological traits and yield” carried out the evaluation of 15 lines of peanuts at the pod filling stage against waterlogging as the stress factor. They further carried measurements of various biochemical processes, such as the concentration levels of the antioxidants such as the CAT, SOD in addition to other byproducts of oxidative stress such as the MDA. What came out in their findings is that the concentration levels of the WTC of REL, MDA content, SOD and CAT were decreased compared with that after waterlogging. This could not be true, a plant under stress will increase the production level of MDA as pointed out in various work done by Magwanga et al (2018, 2019, 2020 and 2021), so this anomaly must be corrected.
Introduction
The introduction is unnecessarily long, with information best transferred to either discussion or methodology, a number of the section describes why the use of a particular parameter, and yet these information’s are the best fit under the materials and methods. Paragraphs 3 to 5 needs to be summarized and much of their contents are either transferred to materials and methods or discussion.
Material and methods
Plant materials: the authors need to justify why the 15 lines were chosen, this will help the readers to understand the reason for this research work. Are they the best performing lines or potentially tolerant to waterlogging. This information needs to be made clear to the readers.
Waterlogging treatment: this section is highly mixed up, let the authors stick to waterlogging as a treatment and not mix with the data collection. The critical question is that can groundnuts survive in a waterlogged environment for a whole week, I doubt. So proper justification must be provided to the authors. Furthermore, SPAD and other antioxidant evaluation sections 142 to 148 must be removed. Moreover, it is better to state the dates in periods either after two weeks of planting or a month after planting as opposed to indicating the actual dates. This paper will be read beyond this century.
Determination of SPAD value and chlorophyll fluorescence parameters: one wonders what you meant by the term “ functional leaf” or “functional leaves” the nature of peanuts has nodes, so it would be proper to indicate the position of the measured leaf. What was the significance of taking the SPAD values in addition to the Chlorophyll fluorescence parameter??
Determination of the activities of SOD and CAT: SOD was evaluated as per the previous method though with modification, it would be proper to indicate the modifications, this will help in the repeatability of this particular research. Moreover, you have gone to write the whole process, so it doesn’t warrant the need to indicate the SOD measurements were as per the previous method.
Results
Before carrying out correlation, it would be good to carry out ANOVA of the various trait measured among the 15 accessions in this experiment. Once ANOVA is done, then the authors can do correlation analysis.
The authors need to avoid the term “As can be seen from Table 2” and consider replacing it with as shown or indicated
The results section is not well written, and this could be attributed to the very basic analysis; in the introduction, several methods were indicated such as “Membership function analysis, principal component analysis and cluster analysis” the cluster analysis helps in determining the hotspots while PCA applicable in data reduction. Can the authors help the readers to understand why these were mentioned and not shown under results?
Discussion
In this section, misrepresentation of facts has been done “Waterlogging stress can not only damage cell membranes but also reduce the photosynthetic capacity of plant leaves. Studies showed that with the prolongation of waterlogging time, MDA content in leaves or roots increased, and the activities of SOD and CAT increased (Zhang et al.,2015; Tian et al., 2019)”
The language of expression is wanting “decreased gradually with the passage of experiment time” and I strongly recommend the authors to redo this section in line with properly analysed data.

Reviewer 2 ·

Basic reporting

The manuscript entitled with "Waterlogging tolerance and recovery capability screening in peanut: a comparative analysis of waterlogging effects on physiological traits and yield" reported the changes of enzyme activities and fluorescence parameters among 15 peanut varieties at the pod filling stage after 7 days of waterlogging and drainage. Basically, the manuscript is not acceptable due to the innovation. Why the authors study the waterlogging stress in peanut? Are they wterlogging sensitive? Why waterlogging for 7 days? I cannot get a clear conclusion after reading the manuscript. Data of the manuscript looks just a section for some other studies.

Experimental design

Why choosing the 7 days of waterlogging and 7 days of drainage? The seeds were sown on 15 August 2020, where are the three replicates to exclude the environment factors?

Validity of the findings

I strongly suggest the authors provided a figure showing the differences of these varieties upon waterlogging and the following recovery stages. The discussion section should relate your study findings to those of other studies.

·

Basic reporting

No comment

Experimental design

No comment

Validity of the findings

No comment

Additional comments

This paper presented a new method to evaluate the waterlogging tolerance of peanut, which has a certain guiding significance for waterlogging tolerant breeding and high yield cultivation of peanut. Based on the comprehensive consideration of the antioxidant and photosynthetic characteristics of leaves after 7 days of waterlogging and 7 days of drainage, the author judged the waterlogging tolerance and recovery of different peanut varieties, and explained the relationship between waterlogging tolerance and recovery and yield. The topic of this paper meets the requirements of the journal, the experiments have certain scientific significance, the experiment design is reasonable, and the logic of context writing is clear. And the diagrams and tables conform to the specification.
In the discussion section, in-depth discussion was carried out, but the deficiency lies in the grammatical problems in the description and expression of the content, and sometimes the narration was too long-winded. Part of the description was not detailed and specific enough.

Introduction:
In the Introduction section, the author can well show the current research progress, research gaps, and put forward the purpose of this experiment.

Materials and methods, and results
The Materials and methods section had concrete details and information, and the author developed a complete description according to the experimental results. But in the waterlogging treatment section, the deficiency was that there was no concrete explanation for "the blank control group".

Discussions:
In the Discussion section, the author can put forward the main findings of this study, which can be summarized as follows: a) different peanut varieties have different responses to waterlogging stress; b) there was no significant relationship between waterlogging tolerance and recovery capability of different varieties, but there was a significant negative correlation between recovery capability and yield decline; c) It is an effective breeding method to select varieties with high recovery capability and less pod number reduction under waterlogging stress.
However, some language descriptions were wordy, and there were grammatical issues. Therefore, it is necessary to make some modifications to the Discussion section, especially to simplify the language.

1. It is necessary to carry out in-depth discussion around three points: a) different peanut varieties have different response mechanisms to waterlogging stress; b) the relationship between waterlogging stress, recovery capability, and yield and yield components; c) The recovery capability of peanut is not the only factor that determines the final yield. In waterlogging tolerance breeding, it is necessary to consider the number of peanut pods per plant after waterlogging.
2. It is not advisable to simply list references, such as L409-417, L425-429, which should be discussed in combination with the topics discussed.
3. The descriptions of some contents are repetitive, such as L373-376 and L412-414. Please simplify the contents.

Other minor mistakes:
L20: Please delete “and” in this sentence.
L30: Please check the grammar of this sentence.
L34: Using “Waterlogging tolerance” and “Recovery capability” is more appropriate to the topic.
L45: To make the sentence more concise, deleting "the process of" is necessary.
L49: Please change “leading to” to “resulting in”.
L50: Delete “resulting in”.
L54: Add “the” before “photosynthetic system”.
L57: Change “sensitive” to “susceptible”.
L63-70: The language should be improved and please check the grammar.
L84-87: The description is wordy.
L91: Please change “led to” to “resulted in”.
L92-94: Please check the grammar of this sentence.
L108-113: It is not enough to list references. And arguments or insights on the subject are required.
L120: Please delete “the” in this line.
L142-144: “SPAD value and chlorophyll fluorescence parameter of leaves were measured after 7 days of waterlogging and 7 days of drainage” should be revised to “SPAD values and chlorophyll fluorescence parameters of leaves were measured after 7 days of waterlogging and 7 days of drainage”.
L169: Please change “record” to “recording”.
L185: Please check the grammar of this sentence.
L228: Please delete “the” in this line.
L238: Please add the sentence “Waterlogging significantly affected the photosynthetic characteristics and antioxidant capacity of peanut leaves. However, there were significant differences in the response of different varieties to waterlogging. ” at the beginning of the paragraph.
L248-250: Please delete the sentence “Waterlogging significantly affected the photosynthetic characteristics and antioxidant capacity of peanut leaves. However, there were significant differences in the response of different varieties to waterlogging” at the beginning of the paragraph.
L252-253: The original content is repeated, changing this to "indicating that waterlogging had a negative impact on yield and yield components" is more appropriate.
L258-263: This paragraph can accurately express the deficiency of correlation analysis and the necessity of principal component analysis, but it is better to use a more concise expression.
L285 and L306: Please change “ranks” to “ranked”.
L287: The use of the word 'worst' is inaccurate.
L320 and L322: Please delete “the” in this line.
L346: “ It shows that the recovery ability of peanut affects peanut yield.” should be changed to “It showed that the recovery ability of peanut affected the final pod yield”.
L363-365: The sentence “Previous studies have shown that the most sensitive growth stage of peanut to waterlogging is the pod filling stage. Waterlogging can lead to peanut rotten pods and finally lead to yield penalty”, please check the grammar and simplify this sentence.
L370-371: “Waterlogging stress can not only damage cell membranes, but also reduce the photosynthetic capacity of plant leaves” should be corrected to “Waterlogging stress damaged cell membranes and reduced the photosynthetic capacity of plant leaves”.
L376: Change “are” to “were” is necessary.
L378: Delete the “antioxidant enzyme”.
L379-381: “And in this study, most of the peanut varieties evaluated showed similar patterns in physiological responses to waterlogging stress, which is consistent with the results of previous studies”, the description of this sentence is abstract, and it is better to delete it.
L386 and L390: Please delete “the” in these lines.
L423-424: Please check the grammar of this sentence and revise it.
L431: Please delete change “are” to “was”.
L432: Please delete “the” in this line.
L454-456: The sentence “Plant stress tolerance is a physiological response affected by many factors and comprehensively manifested by a variety of indicators. Different varieties also have different responses on a single index” is wordy.

In the “Note” of Table 9, the "p" should be in italics.

---

## Round 0.2 · Minor Revisions

Please revises the manuscript according to the reviewers' comments.

·

Basic reporting

The manuscript by Zeng et al, have done an interesting study on evaluating the effects of waterlogging on physiological and yield component in peanut. The information presented is clear and the literature cited is relevant.

Experimental design

The experimental design is appropriate, except the authors need to justify the choice of the 15 cultivars. Moreover, how the seeds used for this study was developed wasn't necessary, more so line 100 to line 104.

Validity of the findings

The results as presented is OK.

Additional comments

I do recommend minor corrections

·

Basic reporting

Dear editor:
The paper entitled “Waterlogging tolerance and recovery capability screening in peanut: a comparative analysis of waterlogging effects on physiological traits and yield (#64480)” evaluated the waterlogging tolerance and recovery capability of different peanut varieties by measuring physiological indexes of peanut leaves after waterlogging and recovery, and established a comprehensive evaluation method of peanut waterlogging tolerance. This paper clearly stated that cultivars with high recovery capability after waterlogging and less pod number reduction should be selected during cultivation. The results and conclusions are well stated, linked to original research questions, which laid a foundation for peanut waterlogging-tolerant varieties breeding.
The revision of the original manuscript generally meets the requirements, and the author has made modifications to the existing grammatical problems. The Discussion section also highlights the research problems and results.
The submission has been greatly improved and is worthy of publication.

Experimental design

No comments.

Validity of the findings

no comment

---

## Round 0.3 · accepted · Accept

The manuscript is properly revised and acceptable.